# Caregiving Intensity, Duration, and Subjective Financial Well-Being Among Rural Informal Caregivers of Older Adults with Chronic Illnesses or Disabilities

**DOI:** 10.3390/healthcare12222260

**Published:** 2024-11-13

**Authors:** Sampada Wagle, Siqi Yang, Evans Appiah Osei, Bhagyashree Katare, Nasreen Lalani

**Affiliations:** 1Department of Agricultural Economics, Purdue University, West Lafayette, IN 47907, USA; wagle0@purdue.edu (S.W.); bkatare@purdue.edu (B.K.); 2Department of Public Health, Purdue University, West Lafayette, IN 47907, USA; yang1703@purdue.edu; 3School of Nursing, Purdue University, West Lafayette, IN 47907, USA; eosei@purdue.edu

**Keywords:** financial burden, informal caregivers, rural, well-being, caregiving intensity

## Abstract

Introduction: Rural informal caregivers (IC) experience major financial and economic constraints in caring for their older family members. Rurality combined with increased caregiving demands and intensity, poor economic opportunities, and limited financial resources and policies create multiple financial stressors and can lead to poor financial well-being. A cross-sectional survey was conducted to understand how caregiving demands, intensity, and duration impact the subjective financial well-being of rural caregivers of older adults. Methods: Informal caregivers (N = 196) residing in 12 rural counties in the central North Region of the Midwestern US participated in the survey. Ordinary Least Squares and Linear Probability Model regressions were conducted to measure the association among the study variables. Results: Our findings showed a moderate level of subjective financial well-being among informal caregivers (average = 51.62; SD 14.52). Caregiving intensity negatively affected financial well-being (β = −1.470, *p* < 0.05). More than half of informal caregivers (58%) were not satisfied with their household income, and 30% found it difficult to meet their family’s needs with their current income status. Discussion and Conclusions: Longer hours of care are associated with financial burden and insecurity and can significantly influence the financial health and well-being of rural informal caregivers of older adults. Older caregivers were found to manage their financial constraints more effectively. Future comparative and longitudinal studies with a more diverse sample are required to infer long-term interactions among the different variables in this study.

## 1. Introduction

About 41.8 million caregivers provide unpaid care to recipients aged 50+ [1]. The economic value of unpaid family care is USD 600 billion in 2021 [2]. By 2030, it is projected that 73 million Americans 65 years of age or older will require daily assistance [3]. The increasing longevity, prolonged disability periods, and rising healthcare costs have burdened informal caregivers [4]. About one in six caregivers of recipients aged 50+ report experiencing high financial strain [1]. The economic impact of unpaid family caregiving is substantial, with an estimate of the opportunity cost as high as USD 67 billion in lost earnings [5]. Caregivers spend a substantial portion of their income, short-term savings, and long-term savings while taking care of their older family members at home [1]. Policies such as the Balanced Incentives Program (BIP) under the Patient Protection and Affordable Care Act were established to allocate additional funds for caregivers and encourage caregiving in the home and community-based areas [6]. However, many who did not receive that funding have experienced increased financial pressures. These disparities underscore the need to examine the financial implications of caregiving among informal caregivers.

Rural caregivers face additional financial threats and insecurities due to several structural inequities, lack of resources, and difficulty in accessing affordable services for their recipients, as compared to their suburban or urban counterparts [1,7,8]. The changing rural demographics, family structures, economic opportunities, and migration trends bring additional challenges and financial insecurity to rural communities. Caregivers, especially those who are employed, carry a dual burden of caregiving. Increased caregiving hours, distance from the care recipient, and the need to miss work or reduce working hours put a huge financial strain on employed caregivers [1,9,10,11,12]. Similarly, early initiation of caregiving and gender roles has also been associated with increased financial vulnerability [13,14,15]. Schulz et al. [16] have identified that the extensive time commitment required to care for a sick, elderly relative can be equivalent to a full-time job. Consequently, caregivers have frequently faced job loss, reduced economic opportunities, withdrawal from the labor market, or reduced working hours, all of which significantly affect their income, expenditures, and financial security [17,18,19,20,21]. Moreover, caregivers also face enormous challenges in coordinating care, exploring care options, identifying services, understanding eligibility, making crucial decisions about care provision, and interacting with healthcare professionals and insurers [22]. Given the above, there is an urgent need to examine the complex interplay between caregiving roles and demands and its impact on the financial well-being of the rural caregivers of older family members in order to inform best practices and policies for improving older adults' and caregivers’ health and well-being outcomes.

Behavioral finance argues that human psychology plays a significant role in shaping financial decisions, leading to behaviors that may not align with rational economic theories [23]. Hashmi et al. [24] have found optimism, impulsivity, and a sense of control affect an individual’s financial behaviors and skills. Netemeyer et al. [25] have found subjective financial well-being to be a major predictor of a person’s general well-being, suggesting a strong link between financial health, happiness, and life satisfaction. Based on the above literature, our study defines subjective financial well-being as an individual’s evaluation of their financial situation, including perception of security, satisfaction, and overall financial health. Our study objectives are (1) to explore the subjective financial well-being of the rural informal caregivers of older adults with chronic illnesses and disabilities and (2) to examine the effects of caregiving roles and demands on caregivers’ finances and how they vary across socio-demographic and financial characteristics.

## 2. Methods

We used a cross-sectional survey approach to gather data about the effects of caregiving on the financial well-being of rural caregivers. A Qualtrics-based survey was developed and administered in twelve North Central rural counties in the US, namely Illinois, Indiana, Iowa, Kansas, Michigan, Minnesota, Missouri, Nebraska, North Dakota, Ohio, South Dakota, and Wisconsin. The survey took about 15–20 mins to complete. A total of (N = 196) participants completed the surveys from December 2023 to March 2024. Participants’ eligibility criteria included the following: 18 years or older, providing informal care to an older adult (60 years or above) with chronic illnesses or disabilities at home, hospice, or residential care setting.

### 2.1. Survey Measures

Dependent Variables: The primary outcome of subjective financial well-being was assessed using three variables: self-reported financial well-being score, feelings about family income, and income sufficiency. To generate a self-reported financial well-being score, we used an abbreviated five-item version of the Consumer Financial Protection Bureau’s (CFPB) financial well-being scale. It comprises a set of five-point Likert scale questions to assess caregivers’ subjective financial health. The scale consisted of three items assessing the perception of financial health: (1) ‘Because of my money situation, I feel like I will never have the things I want in life’; (2) ‘I am just getting by financially’; and (3) ‘I am concerned that the money I have or will save won’t last.’ Similarly, two items explaining individual financial health: (1) I have money left over at the end of the month, and (2) My finances control my life were assessed. We summed the responses in the five items to generate a cumulative financial health outcome. The scale adjusts the final score based on the age of the respondent and the way the survey was administered. The age was dichotomized as 18–61 years and older than 62 years. The way of administering the survey was categorized as self-administered and administered by someone else. Higher scores indicate higher financial well-being [26]. CFPB financial well-being scale has a Cronbach’s alpha of 0.805 in this study.

Perceptions about family income were measured as “How do you feel about your family income?” on a seven-point scale, also known as the delighted/terrible scale devised by Andrews and Withey [27]. The scores ranged from (1) terrible, (2) unhappy, (3) mostly dissatisfied, (4) mixed, (5) mostly satisfied, (6) pleased, and (7) delighted). Responses were converted into a binary variable: (0) unsatisfied (including terrible, unhappy, mostly dissatisfied, and mixed) and (1) satisfied (including mostly satisfied, pleased, and delighted).

Income sufficiency was assessed using a single item in the survey questionnaire: “After tax, how would you consider your income to provide for yourself and/or your family?” Responses ranged from insufficient, sufficient, good, and very good. Responses were calculated as a dichotomous measure: (1) sufficient (including sufficient, good, and very good) and 0 insufficient.

Independent Variables: The primary independent variables included caregiver intensity and duration, which were analyzed as continuous variables. Caregiving intensity represents the average number of hours caregivers spend on caregiving in a week. For caregiving intensity, participants were asked, ‘On average, how many hours of a week do you provide care to this person?’ Similarly, caregiving duration indicates the total number of months that caregivers have reported they have been providing care, representing the length of time they have taken on caregiving responsibilities. For the duration of caregiving, participants were asked, ‘How many months/years have you been providing care to the care recipient?’.

Control Variables: Potential covariates were based on the prior literature and included caregiver/care recipient characteristics such as the location of the care provided, relationship with the care recipient, proximity to the care recipient, and type of illness/disability of the care recipient. Other demographic variables included age, gender, race/ethnicity, education level, employment status, marital status, number of adults and children in the house, and household income.

### 2.2. Data Analysis

We used STATA version 18.0 for data analysis. Bivariate Pearson’s correlations were carried out to examine the association among the caregiving characteristics and three financial well-being outcome variables. We employed Ordinary Least Squares (OLS) regression for self-reported financial well-being outcomes, whereas we used Linear Probability Model (LPM) for binary outcomes related to feelings about family income and income sufficiency.

### 2.3. Ethical Considerations

IRB was obtained from the University IRB committee. Online consent was obtained for the surveys embedded in the Qualtrics form. Anonymity and confidentiality were maintained following the ethical guidelines. Surveys were number-coded and did not include any personal identifiers. All the survey data were stored on Qualtrics and shared only with the research team.

## 3. Results

### 3.1. Sample Characteristics

The demographic characteristics of the survey participants are shown in Table 1. The average age of the caregiver was 50 years (SD = 16.80). On average, a caregiver spent 15 h (SD = 21.75) providing care for 25.71 months (SD = 33.55). Over half (52%) of the caregivers provided care to their parents or parents-in-law. Nearly (61%) of them were providing care in the care recipient’s home, and 71% of caregivers were found living close to their care recipients. Almost 87% of caregivers attended to chronically ill individuals, highlighting that chronic illnesses were a common reason for long-term care. Caregivers experienced a moderate level of financial well-being (mean scores = 51.62). Less than half (42%) were satisfied with their family income, while most (70%) felt their earnings met their family’s needs. Notably, not all who perceived their income as sufficient were satisfied.

### 3.2. Associations Among Study Variables

Bivariate correlations for the study variables (Table 2) showed that three financial well-being outcomes were positively correlated. Caregivers’ self-reported financial well-being correlated significantly with feelings of satisfaction with their family income (r = 0.5439, *p* < 0.01) and income sufficiency (r = 0.5248, *p* < 0.01). The caregivers who perceived high financial well-being were likely to be more satisfied and felt their household income to be sufficient. Similarly, feelings about family income and income sufficiency were positively correlated (r = 0.4592), suggesting that those who found their income to be sufficient were likely to be more satisfied. All these findings indicate that the various aspects of subjective financial well-being are likely interconnected, and improvements in one aspect could positively affect the others.

Table 3 depicts the coefficients of the OLS and LPM regression models. The F-test results were significant for all models, indicating that the predictors explained the variation in subjective financial well-being outcomes. The R-squared values indicated that the models explained a moderate proportion of the variance in the subjective financial well-being outcomes, with the highest explanatory power for self-reported financial well-being.

The OLS regression (Column 1) revealed several significant predictors of self-reported financial well-being among informal caregivers. Caregiving intensity measured by average hours of care provided per week negatively affected financial well-being (β = −1.470, *p* < 0.05), indicating that for every 1% increase in the average hours of care provided per week, there is an expected decrease in financial well-being by approximately 0.01470 units. Caregiving duration, however, did not show a significant impact on the financial well-being outcomes of the caregivers. Among the demographics, the age of caregivers was positively related to financial well-being, indicating that older caregivers experienced slightly higher financial well-being (β = 0.155, *p* < 0.05). Race and gender showed no significant association. There was a strong association with education (β = 5.467, *p* < 0.01), indicating caregivers who graduated with a degree after a high school diploma experienced better financial well-being. Married caregivers had better financial health (β = 4.631, *p* < 0.05). The employment status and the number of children in the household were associated with lower financial well-being (β = −3.845, *p* < 0.10; β = −2.748, *p* < 0.05, respectively).

Similarly, LPM regression (Columns 2 and 3) showed that caregiving factors, including caregiving intensity and duration, generally did not significantly affect the caregivers’ perceived feelings of income satisfaction and income sufficiency. However, care provided in the care recipient’s house was associated with a decreased probability of the caregivers’ perceiving their income as sufficient (β = −0.125, *p* < 0.10). Higher education (β = 0.165, *p* < 0.05) and higher household income (β = 0.198, *p* < 0.05) significantly influenced the caregiver’s probability of feeling satisfied with their family income. Employed caregivers were likely to be less satisfied with their household income (β = −0.133, *p* < 0.10). Caregivers with higher education and higher household incomes were more likely to perceive their income as sufficient to meet their family needs (β = 0.127, *p* < 0.10; β = 0.163, *p* < 0.05, respectively).

## 4. Discussion

This study examined the effect of caregiving roles and demands on the financial well-being of rural informal caregivers caring for their older family members with chronic illness or disability. We found a strong relationship between caregiving intensity and the financial well-being of informal caregivers of older adults. Financial burden and insecurity increased with the longer hours of care provided in a week and significantly affected the financial health and well-being of rural caregivers. Similar findings have also been reported in previous studies where caregivers had to reduce their working hours or leave their jobs and businesses, resulting in financial strains and insecurity [1,28,29,30]. Our analysis revealed no significant relation between the duration of caregiving and subjective financial well-being outcomes, indicating the total length of time caregivers spent providing care does not directly affect the perceptions of their financial well-being. Together, our findings show that the intensity of caregiving (measured in hours per week) significantly reduces perceived financial well-being, while the total duration of caregiving (measured in months) has no effect on perceived financial well-being. Future research with additional variables may be conducted to better explain this phenomenon of caregivers’ subjective financial well-being. It is significant to note in our study that all three financial well-being outcomes—including perceived financial well-being, feelings of satisfaction about family income, and feelings of the sufficiency of family income—were positively and significantly correlated with each other. This highlights the interconnected nature of financial health indicators and the potential for comprehensive strategies to improve the overall financial well-being of rural caregivers.

Our findings suggested that informal caregivers perceive lower income sufficiency when providing care at the care recipient’s home. One possible explanation could be the additional responsibilities and financial expenses incurred by caregivers in this setting. These expenses can be related to transportation, medical and homemaking equipment and supplies, and prescription and nonprescription medications, which can collectively strain the caregivers’ budget, affecting financial adequacy and perceived financial well-being [31,32]. This finding aligns with previous research [1,33], which has shown that caregivers who live with their care recipient are more likely to report increased financial strain compared to those who do not co-reside. Other caregiving characteristics, such as relationship with the care recipient, proximity to the care recipient, and type of illness/disability of the care recipient, did not show a significant relationship with subjective financial well-being. Age was found to be a significant factor in assessing the financial well-being of caregivers. Prior studies have suggested that older caregivers may face more economic difficulties [15,34]. On the contrary, our study showed older caregivers had better financial outcomes. This may be likely as older caregivers may have accumulated retirement savings and may be eligible for Social Security or pension benefits, providing a buffer against the costs of caregiving. This financial security may give older caregivers more freedom to manage their work and caregiving responsibilities. Young caregivers, on the other hand, frequently have smaller retirement contributions and savings, which leaves them more susceptible to the long-term financial effects of fewer hours worked or job disruptions, thereby raising the expenses of providing care [35]. A recent AARP report also demonstrated that the financial impacts of caregiving were more pronounced in younger caregivers, who often struggled to afford necessities, pay bills, and borrow money from family and friends [1].

Similarly, marital status affected the financial well-being of caregivers, with married caregivers reporting better financial well-being. Combined household income and shared resources among partners support caregiving roles and abilities, thereby reducing financial stress [36,37]. Similarly, the size of the family negatively influenced the financial well-being. It aligns with the findings of Kim and Waite (2013) [38], who have shown that the increased number of family members increases financial demands, making it more challenging to distribute and allocate resources within the family.

Previous studies have shown that women typically face a higher financial burden due to factors like lower earnings, social security, pension benefits, and increased caregiving-related expenses; our study found no significant effect of gender on financial well-being outcomes [14]. Similarly, non-white or minority caregivers are more likely to report increased financial strain [1,39]. However, in our study, we could not observe financial well-being disparities across racial groups, as most of the participants were white.

Higher education is associated with better financial outcomes for caregivers. This finding is consistent with the recent AARP report, where caregivers with no college degree reported an increased financial burden [1]. Higher education may bring better employment opportunities and increase the ability to effectively manage financial resources [40]. Our study showed that employment status negatively affected perceived financial well-being. Prior studies have found that employment can be both a source of financial support and a stressor due to the nature and conflicting demands of work and caregiving roles and responsibilities [29,30].

Our study informs several policy implications for promoting the financial well-being of rural caregivers. Future policies and programs should consider caregiving intensity, location, hours of caregiving, educational background, marital status, and family dynamics. Financial well-being should be seen beyond monetary or income perspectives. Higher education and better employment opportunities do not guarantee financial well-being and may influence caregiving demands and financial satisfaction in different ways. Caregiver support programs should include topics around financial literacy and resources to assist caregivers in making better financial choices and decision-making to ensure better caregiving practices, aging, and health outcomes in rural areas.

## 5. Conclusions

Our study highlights the need for and importance of addressing caregiving intensity and its impact on the financial well-being of informal caregivers of older family members in rural settings. Rural caregivers providing longer hours of caregiving for their older family members at home were found to be at increased risk of financial strain and poor financial well-being, indicating the need for tailored caregiver support interventions to promote overall aging, health, and well-being outcomes in rural communities.

## 6. Limitations and Strengths

This study brings attention to the unique financial stressors faced by caregivers in rural settings with a robust sample size. However, the sample is limited to informal caregivers in twelve rural counties in the North Central region of the Midwestern US. As a result, the findings may not be generalizable to caregivers in other rural or urban areas or to diverse populations with different socioeconomic backgrounds. This study relies on self-reported survey data, which may introduce recall bias or social desirability bias, where participants might underreport or overreport their financial stress or caregiving experiences. Future comparative and longitudinal studies with a more diverse sample are required to infer long-term interactions among the different variables in this study.

## Figures and Tables

**Table 1 healthcare-12-02260-t001:** Sample Characteristics (N = 196).

Self-reported financial well-being (mean, s.d.)	51.62 ± 14.52
Feelings about family income (1 = Satisfied) (%)	82 (41.84%)
Income sufficiency (1 = Sufficient) (%)	138 (70.41%)
Average hours of care provided per week (mean, s.d.)	15.096 ± 21.756
Average hours of care provided per week (log-transformed) (mean, s.d.)	1.927 ± 1.266
Length of care provided (months) (mean, s.d.)	25.71 ± 33.55
Age (mean, s.d.)	49.98 ± 16.80
Gender (1 = female) (%)	104 (53.06%)
Race (1 = white) (%)	160 (81.63%)
Education (1 = graduated with a degree after high school diploma) (%)	95 (48.47%)
Relation status (1 = Paired) (%)	121 (61.73%)
Employment status (1 = Employed) (%)	97 (49.49%)
Number of adults in the house (mean, s.d.)	1.53 ± 1.12
Number of children in the house (mean, s.d.)	0.50 ± 0.88
Household income (1 = high-income (>=50,000)) (%)	101(51.53%)
Relation with older adults (1 = Parent/Parent-in-laws) (%)	102 (52.04%)
Location of the care (1 = Older adults’ home) (%)	119 (60.71%)
Location difference between caregivers and older adults (1 = same town/city as older adults) (%)	140 (71.43%)
Type of illness/disability (1 = Chronic) (%)	170 (86.73%)

Notes: Mean (standard deviations) are reported for continuous variables. Percentages reported for binary variables.

**Table 2 healthcare-12-02260-t002:** Bivariate correlations among caregiving factors and financial well-being outcomes (N = 196).

		Correlation Matrix				
		1	2	3	4	5
1	Average hours of care provided per week	--------				
2	Length of care provided	0.0901	--------			
3	Self-reported financial well-being	−0.0973	0.0601	--------		
4	Feelings about family income	0.0092	0.0687	0.5439 ***	-----------	
5	Income sufficiency	−0.0359	0.0559	0.5248 ***	0.4592 ***	--------

Notes: *** *p* ≤ 0.01. Diagonal cells (marked with dashes) represent self-correlations, which are always equal to 1, and are therefore omitted for clarity.

**Table 3 healthcare-12-02260-t003:** Regression estimates for financial well-being outcomes (N = 196).

	Financial Well-Being	Feel About Family Income	Income Sufficiency
Average hours of care per week (log-transformed)	−1.470 ** (0.712)	−0.001 (0.028)	−0.008 (0.027)
Length of care provided (months)	0.004 (0.029)	0.0005 (0.0010)	0.0001 (0.0010)
Gender (1 = female)	−0.984 (1.978)	0.074 (0.078)	−0.088 (0.068)
Age of caregivers	0.155 ** (0.064)	0.0006 (0.0028)	0.002 (0.002)
Education (1 = graduated with a degree after high school)	5.467 *** (1.917)	0.165 ** (0.074)	0.127 * (0.069)
Race (1 = white)	−3.098 (2.968)	0.041 (0.090)	−0.062 (0.099)
Marital status (1 = married)	4.631 ** (2.352)	0.050 (0.083)	0.030 (0.087)
Number of children in the house	−2.748 ** (1.239)	−0.011 (0.042)	−0.016 (0.041)
Employment status (1 = employed)	−3.845 * (2.129)	−0.133 * (0.079)	−0.080 (0.076)
Household income (1 = high-income (>=50,000)	2.287 (2.298)	0.198 ** (0.084)	0.163 ** (0.082)
Location of care provided (1 = care recipient’s home)	−2.370 (2.019)	−0.012 (0.075)	−0.125 * (0.073)
R^2^	0.2498	0.1418	0.1438

Notes. The first column represents the results of the OLS regression model, while the second and third columns represent the results of the LPM regression model. Standard errors (in parentheses) are corrected for heteroscedasticity. Specification controls for the number of adults in the household, relation with care recipients, proximity to care recipients, and type of illness or disability. * *p* ≤ 0.10, ** *p* ≤ 0.05, *** *p* ≤ 0.01.

## Data Availability

The original contributions presented in this study are included in the article; further inquiries can be directed to the corresponding author.

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
