# Peer review of "Caregiving Intensity, Duration, and Subjective Financial Well-Being Among Rural Informal Caregivers of Older Adults with Chronic Illnesses or Disabilities"

_healthcare, 2024, doi:10.3390/healthcare12222260_

Round 1

Reviewer 1 Report

Comments and Suggestions for Authors

Review of “Caregiving Intensity, Duration, and Subjective Financial Well-Being of Rural Informal Caregivers of Older Adults with Chronic Illnesses or Disabilities”

Summary

The paper investigates the subjective financial well-being of rural informal caregivers who provide care for older adults with chronic illnesses or disabilities, as well as its relationship with caregiving intensity and duration. The study is based on cross-sectional survey data collected from 196 participants across twelve rural counties in the Midwestern United States.

Primary Comments

The topic is both interesting and important, and the paper is well-written. My main comment is outlined below.

As the title suggests, the primary aim is to explore the relationship between subjective financial well-being and the intensity and duration of caregiving. However, the overall relationship is weak, particularly with caregiving duration, where no significant results were found. The only significant correlation identified is between caregiving intensity and one of the three financial well-being variables. While the absence of a relationship is still a finding, the authors should provide more detailed discussion on the results and their implications.

Secondary Comments

The authors mentioned in the abstract that "Older caregivers were found to manage their financial constraints using their retirement savings." However, there is no supporting evidence from the analysis to back this claim. At best, it is a conjecture based on the finding that "the age of caregivers was positively related to financial well-being."

Author Response

Dear Reviewer 1,

Response to Reviewer 1 Comments

1. Summary

2. Questions for General Evaluation

Reviewer’s Evaluation

Response and Revisions

Does the introduction provide sufficient background and include all relevant references?

Yes

[Thank you]

Are all the cited references relevant to the research?

Yes

[Thank you]

Is the research design appropriate?

Yes

[Thank you]

Are the methods adequately described?

Yes

[Thank you]

Are the results clearly presented?

Yes

[Thank you]

Are the conclusions supported by the results?

Can be improved

[Thank you for the comment, the conclusion has been revised and highlighted]

3. Point-by-point response to Comments and Suggestions for Authors

Comments 1: [Primary Comments]

The topic is both interesting and important, and the paper is well-written. My main comment is outlined below.

As the title suggests, the primary aim is to explore the relationship between subjective financial well-being and the intensity and duration of caregiving. However, the overall relationship is weak, particularly with caregiving duration, where no significant results were found. The only significant correlation identified is between caregiving intensity and one of the three financial well-being variables. While the absence of a relationship is still a finding, the authors should provide more detailed discussion on the results and their implications.]

Response 1:

Comment have been addressed. Findings have been discussed as: Our analysis revealed no significant relation between the duration of caregiving and subjective financial wellbeing outcomes. It suggests that the quantity of time caregivers spend providing care does not appear to have a direct effect on the perceptions of their financial wellbeing. These results underscore the necessity of more research on additional variables that could impact caregivers' subjective financial well-being.

Comments 2: Secondary Comments

The authors mentioned in the abstract that "Older caregivers were found to manage their financial constraints using their retirement savings." However, there is no supporting evidence from the analysis to back this claim. At best, it is a conjecture based on the finding that "the age of caregivers was positively related to financial well-being."

Response 2:

It has been revised as: Older caregivers were found to manage their financial constraints more effectively.

Reviewer 2 Report

Comments and Suggestions for Authors

Review report for Caregiving Intensity, Duration, and Subjective Financial Well-Being of Rural Informal Caregivers of Older Adults with  Chronic Illnesses or Disabilities

Abstract

Research problem and research objective are not clearly conveyed to the readers.

Literature review

Literature review section is missing. What are the theories that drive the development of this study?

Where is the conceptual framework?

Which studies that support the constructs that you develop in this study?

Results

Where is the Reliability Analysis?

“We did not find any significant differences in financial well-being across genders.” Shouldn’t it be supported by Comparative Analysis?

Conclusion

The statement includes both the positive aspect of sample size and its limitation in generalizability. Instead of emphasizing a "robust sample size" only to state its limitations in the next sentence, clarify whether the sample size genuinely supports the claims and provide precise numbers (e.g., total participants, demographic distribution) to justify robustness. The finding that “older and married caregivers, as well as those with higher education, reported better financial outcomes” may require further explanation to avoid ambiguity. Older caregivers may have additional savings, or higher education may correlate with better-paying jobs; the conclusion could discuss these factors briefly to reinforce clarity.

Author Response

Dear Reviewer 2,

Response to Reviewer 2 Comments

1. Summary

2. Questions for General Evaluation

Reviewer’s Evaluation

Response and Revisions

Does the introduction provide sufficient background and include all relevant references?

Must be improved

[The introduction covers the primary aim, which focuses on both the formal and informal costs of caregiving, as well as the intensity experienced by caregivers. Could you please clarify what additional details you would like us to include in the introduction?]

Are all the cited references relevant to the research?

Must be improved

[The references have been revised?]

Is the research design appropriate?

Must be improved

[Thank you. However, there was no feedback provided on the design. Could you please specify how the design should be revised?

Are the methods adequately described?

Must be improved

Thank you could you please clarify which section of the methods need improvement?

Are the results clearly presented?

Must be improved

Comments have been addressed

Are the conclusions supported by the results?

Must be improved           

[Comments have been addressed]

3. Point-by-point response to Comments and Suggestions for Authors

Comments 1: Abstract

Research problem and research objective are not clearly conveyed to the readers.

Response: Thank you for bringing this to our attention. We agree with the comment regarding the aim. The research problem has been highlighted in the abstract and the aim has been addressed, and highlighted in the abstract as suggested. Could you please clarify any specific changes or revisions you would like to see regarding the problem? Below are the challenges and the aim of this study for your reference. “Rural informal caregivers (IC) experience major financial and economic constraints in caring for their older family members. Rurality combined with increased caregiving demands and intensity, poor economic opportunities, and limited financial resources and policies create multiple financial stressors and can lead to poor financial well-being. Our study investigated the financial well-being of rural informal caregivers (ICs) who care for older family members, to understand how caregiving demands, intensity, and duration impact caregivers' subjective financial well-being]

Comments 2: Literature review

Literature review section is missing. What are the theories that drive the development of this study?

Where is the conceptual framework?

Which studies that support the constructs that you develop in this study?.]

Response 2: The frameworks guiding the development of this study, along with supporting studies for the constructs, have been cited in the introduction, as the manuscript guidelines did not call for a separate literature review section. Please see the information below and let us know if you have any further questions.

 { Behavioral finance has argued that human psychology plays a significant role in shaping financial decisions, leading to behaviors that may not align with rational economic theories [23]. Hashmi et al.[24] have found optimism, impulsivity, and a sense of control affect an individual's financial behaviors and skills. Netemeyer et al.[25] have found subjective financial well-being to be a major indicator of a person's general well-being, suggesting a strong link between financial health, happiness, and life satisfaction. Based on the above literature, our study defines subjective financial well-being as an individual's evaluation of their financial situation, including perception of security, satisfaction, and overall financial health.”

Comment 3: Results

Where is the Reliability Analysis?

Response 3: It has now been included: CFPB financial wellbeing scale has a Cronbach’s alpha of 0.805 in this study.

To examine the overall reliability of three measures, we did Pearson’s bivariate correlation analysis, the results of which are detailed in the section: 3.2. Associations among Study Variables.

“We did not find any significant differences in financial well-being across genders.” Shouldn’t it be supported by Comparative Analysis?]”

We are not doing comparative analysis. So, we have revised the line accordingly.  Although previous research indicates that women typically face a higher financial burden due to factors like lower earnings, social security, pension benefits, and increased caregiving-related expenses, our study found no significant effect of gender on financial well-being outcomes[12].

4. Comment 4: Conclusion

The statement includes both the positive aspect of sample size and its limitation in generalizability. Instead of emphasizing a "robust sample size" only to state its limitations in the next sentence, clarify whether the sample size genuinely supports the claims and provide precise numbers (e.g., total participants, demographic distribution) to justify robustness. The finding that “older and married caregivers, as well as those with higher education, reported better financial outcomes” may require further explanation to avoid ambiguity. Older caregivers may have additional savings, or higher education may correlate with better-paying jobs; the conclusion could discuss these factors briefly to reinforce clarity.

Response 4: The conclusion section has been fully revised to emphasize the key findings of the study, with the limitations removed and placed in their designated section. Please refer to the revised conclusion below for your review.

This study demonstrates that caregiving intensity has a significant negative impact on the financial well-being of rural informal caregivers, particularly those caring for older adults with chronic illnesses or disabilities. Longer caregiving hours were associated with increased financial strain, especially for caregivers residing with the care recipient. Notably, older, married, and more educated caregivers reported better financial outcomes, suggesting that these factors may provide some protective effects against financial stress. However, despite these findings, many caregivers still struggled with income sufficiency and satisfaction, indicating the need for tailored interventions. Future policies and support programs should focus on reducing the financial burden on rural caregivers, promoting self-care, and offering financial literacy resources to improve their long-term financial well-being. Additionally, more diverse and longitudinal studies are needed to further explore the relationships between caregiving intensity, financial health, and demographic factors”

Thank you for your comment regarding the references to related and previous work. We have tried to include a range of relevant studies that address the main themes of our research. However, if there are specific areas or topics you believe require further exploration or additional references, we would appreciate your guidance.

Round 2

Reviewer 2 Report

Comments and Suggestions for Authors

The authors have made the necessary revision